# Provably convergent quasistatic dynamics for mean-field two-player zero-sum games

**Chao Ma, Lexing Ying**
Department of Mathematics
Stanford University
Stanford, CA 94305, USA
{chaoma,lexing}@stanford.edu

## Abstract

In this paper, we study the problem of finding mixed Nash equilibrium for mean-field two-player zero-sum games. Solving this problem requires optimizing over two probability distributions. We consider a quasistatic Wasserstein gradient flow dynamics in which one probability distribution follows the Wasserstein gradient flow, while the other one is always at the equilibrium. Theoretical analysis are conducted on this dynamics, showing its convergence to the mixed Nash equilibrium under mild conditions. Inspired by the continuous dynamics of probability distributions, we derive a quasistatic Langevin gradient descent method with inner-outer iterations, and test the method on different problems, including training mixture of GANs.

## 1 Introduction

Finding Nash equilibrium has seen many important applications in machine learning, such as generative adversarial networks (GANs) (Goodfellow et al., 2014a) and reinforcement learning (Busoniu et al., 2008). In these problems, pure Nash equilibria are usually search for a function $f(x, y)$. Yet, the problems arising from machine learning are usually nonconvex in $x$ and nonconcave in $y$, in which case pure Nash equilibrium may not exist. And even if it exists, there is no guarantee for any optimization algorithm to find it efficiently. This difficulty is reflected in practice, that compared with simple minimization, machine learning applications involving Nash equilibria usually have more complicated behaviors and more subtle dependence on hyper-parameters. For example, stable and efficient training of GANs requires a number of carefully designed tricks (Gao et al., 2018).

On the other hand, the mixed Nash equilibrium (MNE) is known to exist in much more general settings, e.g. when the strategy spaces are compact and the payoff function is continuous (Glicksberg, 1952). In the mixed Nash equilibrium problem, instead of taking "pure strategies" $x$ and $y$, two "mixed strategies" for $x$ and $y$, in the form of probability distributions, are considered, resulting in the following functional,

$$\int f(x, y)p(x)q(y)dxdy,$$

where $p$ and $q$ are density functions of probability distributions of $x$ and $y$, respectively. Efforts are invested to develop theoretically endorsed algorithms that can efficiently find MNE for high dimensional problems, with applications on the training of mixture of GANs. In Hsieh et al. (2019), a mirror-descent algorithm is proposed and its convergence is proven. In Domingo-Enrich et al. (2020), theoretical analysis and empirical experiments are conducted for a gradient descent-ascent flow under a Wasserstein-Fisher-Rao metric and its particle discretization.

In this paper, we also consider the mixed Nash equilibrium problem, and propose a simple QuasiStatic Wasserstein Gradient Flow (QSWGF) for solving the problem. In our dynamics, we treat $q$ as a component with much faster speed than $p$, hence is always at equilibrium as $p$ moves. With entropy regularization for both $p$ and $q$ (without requirement on the strength of the regularizations), we prove that the QSWGF converges to the unique mixed Nash equilibrium from any initialization (under mild conditions). Furthermore, we show there is a simple way to discretize the QSWGF, regardless of the complexity of the Wasserstein gradient flow of $p$ induced by the fact that $q$ is always

at equilibrium. Concretely, a partition function related with $p$ appears in the QSWGF dynamics, and we find an efficient way to approximate the partition function. By discretizing the QSWGF, we derive a particle dynamics with an inner-outer structure, named the QuasiStatic Langevin Gradient Descent algorithm (QSLGD). In QSLGD, after each iteration of the outer problem (for the $x$ particles), the inner loop conducts sufficient iterations to bring the $y$ particles to equilibrium. Numerical experiments show the effectiveness of QSLGD on synthetic examples and training mixture of GANs. Our method outperforms the vanilla Langevin gradient descent-ascent method when the entropy regularization is weak.

As a summary, our two major contributions are:

1. We propose the quasistatic Wasserstein gradient flow dynamics for mixed Nash equilibrium problems, and show its convergence to the unique Nash equilibrium under weak assumptions. Our result neither requires the entropy regularization to be sufficiently strong, nor assumes the dynamics to converge a priori.

2. We derive a simple while practical quasistatic Langevin gradient descent algorithm by discretizing the quasistatic Wasserstein gradient flow, by finding an efficient way to approximate the partition function appearing in the dynamics of $p$. The proposed algorithm is applied on several problems including training mixtures of GANs.

## 2 RELATED WORK

The mixed Nash equilibrium problem has a long history, with the proof of its existence dates back to Morgenstern & Von Neumann (1953). It draws new attention in recent years, especially in the machine learning community, due to the development of GANs (Goodfellow et al., 2014a) and adversarial training (Goodfellow et al., 2014b). Training mixture of GANs is already discussed in paper (Goodfellow et al., 2014a). Some numerical experiments were conducted in (Arora et al., 2017). In Grnarova et al. (2017), the authors proposed an online learning approach for training mixture of GANs, and proved its effectiveness for semi-shallow GANs (GANs whose discriminator is a shallow neural network). Yet, rigorous theoretical treatment to an algorithm started from (Hsieh et al., 2019), in which a mirror descent method was studied and proven to converge. The implementation of the mirror descent method involves big computational cost that asks for heuristics to alleviate. Later, (Domingo-Enrich et al., 2020) studied more efficient algorithms under a mixture of Wasserstein and Fisher-Rao metrics. Theoretically, the time average of the dynamics' trajectories is shown to converge to the mixed Nash equilibrium. As a comparison, in this work we show the global convergence of the quasistatic Wasserstein gradient flow without the need of taking time average. Meanwhile, the Wasserstein nature of our dynamics makes it easy to implement as well.

The Wasserstein gradient flow in the density space has been explored in previous works. For example, (Wang & Li, 2019) studied the Nesterov's accelerated gradient flows for probability distributions under the Wasserstein metric, and (Arbel et al., 2019) studied practical implementations of the natural gradient method for the Wasserstein metric. Both works focus on minimization problems instead of min-max problems considered in this work. A more related work is Lin et al. (2021b), where a natural gradient based algorithm is proposed for training GANs. Yet, the method still optimizes one generator and one discriminator, searching for pure Nash equilibrium. Another work that derives algorithms for GANs from a Wasserstein perspective is (Lin et al., 2021a).

Another volume of works that studies the Wasserstein gradient flow in the machine learning context is the mean-field analysis of neural networks. This line of works started from two-layer neural networks (Mei et al., 2018; Rotskoff & Vanden-Eijnden, 2018; Chizat & Bach, 2018; Sirignano & Spiliopoulos, 2020), to deep fully-connected networks (Araújo et al., 2019; Sirignano & Spiliopoulos, 2021; Nguyen, 2019; Wojtowytsch et al., 2020), and residual networks (Lu et al., 2020; E et al., 2020). The mean-field formulations treat parameters as probability distributions, and the training dynamics are usually the gradient flow under Wasserstein metric. Attempts to prove convergence of the dynamics to global minima are made (Mei et al., 2018; Chizat & Bach, 2018; Rotskoff et al., 2019), though in the case without entropy regularization a convergence assumption should usually be made a priori.

## 3 THE QUASISTATIC DYNAMICS

We consider the entropy regularized mixed Nash equilibrium problem, which in our case is equivalent with solving the following minimax problem:

$$\min_{p \in \mathcal{P}(\Omega)} \max_{q \in \mathcal{P}(\Omega)} \int_{\Omega \times \Omega} K(x,y)p(x)q(y)dxdy + \beta^{-1}\int_{\Omega} p\log p dx - \beta^{-1}\int_{\Omega} q\log q dy. \quad (1)$$

In (1), $\Omega$ is a compact Riemannian manifold without boundary, and $\mathcal{P}(\Omega)$ is the set of probability distributions on $\Omega$. Since $\Omega$ is compact, any probability distribution in $\mathcal{P}(\Omega)$ naturally has finite moments. Let $E(p,q) = \int_{\Omega \times \Omega} K(x,y)p(dx)q(dy)$, and $S(p) = \int_{\Omega} p\log p dx$ and $S(q) = \int_{\Omega} q\log q dy$ be the (negative) entropy of $p$ and $q$, respectively. Then, the minimax problem (1) can be written in short as

$$\min_{p \in \mathcal{P}(\Omega)} \max_{q \in \mathcal{P}(\Omega)} E(p,q) + \beta^{-1}S(p) - \beta^{-1}S(q). \quad (2)$$

**Remark 1.** *Strictly speaking, in (1) we should distinguish probability distributions and their density function (if exist), and the entropy should also be defined using the Radon-Nikodym derivative with canonical measure. In this paper, since $p$ and $q$ indeed have density functions because of the entropy regularization, we shall abuse the notation by using $p$ and $q$ to represent both probability distributions and their density functions.*

The entropy regularizations in (1) and (2) make the problem strongly convex in $p$ and strongly concave in $q$. Hence, there exists a unique Nash equilibrium for the problem. Such results are shown for example by the following theorem from (Domingo-Enrich et al., 2020).

**Theorem 1.** *(Theorem 4 of (Domingo-Enrich et al., 2020)) Assume $\Omega$ is a compact Polish metric space equipped with canonical Borel measure, and that $K$ is a continuous function on $\Omega \times \Omega$. Then, problem (2) has a unique Nash equilibrium given by the solution of the following fixed-point problem:*

$$p(x) = \frac{1}{Z_p}\exp(-\beta U(x,q)), \quad q(x) = \frac{1}{Z_q}\exp(\beta V(y,p)), \quad (3)$$

*where $Z_p$ and $Z_q$ are normalization constants to make sure $p$ and $q$ are probability distributions, and $U$ and $V$ are defined as*

$$U(x,q) = \frac{\delta E(p,q)}{\delta p}(x) = \int_{\Omega} K(x,y)q(y)dy, \quad V(y,p) = \frac{\delta E(p,q)}{\delta q}(y) = \int_{\Omega} K(x,y)p(x)dx.$$

Considering the efficiency in high-dimensional cases, a natural dynamics of interest to find the Nash equilibrium for (2) is the gradient descent-ascent flow under the Wasserstein metric,

$$\partial_t p_t = \nabla \cdot \left( p_t \nabla (U(x,q_t) + \beta^{-1}\log p_t) \right),$$
$$\partial_t q_t = \nabla \cdot \left( q_t \nabla (-V(y,p_t) + \beta^{-1}\log q_t) \right), \quad (4)$$

because it can be easily discretized into a Langevin gradient descent-ascent method by treating the PDEs as Fokker-Planck equations of SDEs. When $\beta^{-1}$ is sufficiently large, (4) can be proven to converge linearly to the unique MNE of (2) (Eberle et al., 2019). However, when $\beta^{-1}$ is small, whether (4) converges remains open. This hinders the application of (4) because in practice the entropy terms are usually used as regularization and are kept small. (We realize that it is proven in Domingo-Enrich & Bruna (2022) when our work is under review.)

In (4), the dynamics of $p$ and $q$ have the same speed. In this work, instead, we study a quasistatic Wasserstein gradient descent dynamics, which can be understood as a limiting dynamics when the speed of $q$ becomes faster and faster compared with that of $p$. In this case, at any time $t$, we assume $q_t$ reaches at the equilibrium of the maximizing problem instantaneously by fixing $p = p_t$ in (2). That is to say, at any time $t$, $q_t$ is determined by

$$q_t = q[p_t] := \arg \max_{q \in \mathcal{P}(\Omega)} E(p_t, q) - \beta^{-1}S(q). \quad (5)$$

On the other hand, $p_t$ follows the Wasserstein gradient descent flow with $q_t = q[p_t]$ at the equilibrium:

$$\partial_t p_t = \nabla \cdot \left( p_t \nabla \left( \frac{\delta(E(p_t, q[p_t]) - \beta^{-1}S(q[p_t]))}{\delta p_t} + \beta^{-1}\log p_t \right) \right). \quad (6)$$

The following theorem shows $q_t = q[p_t]$ can be explicitly written as a Gibbs distribution depending on $p_t$, and thus the free energy in (6) can be simplified to depend on a partition function related with $p_t$.

**Theorem 2.** *Assume $K$ is continuous on the compact set $\Omega$ and $\beta > 0$. Then, for fixed $p_t$ the maximization problem (5) has a unique solution*

$$q[p_t](y) := \frac{1}{Z_q(p_t)} \exp(\beta V(y, p_t)), \tag{7}$$

*where $Z_q(p)$ is a normalization factor, $Z_q(p) := \int \exp(\beta V(y, p)) dy$. Moreover, the dynamics (6) for $p_t$ can be written as*

$$\partial_t p_t = \nabla \cdot \left( p_t \nabla \left( \frac{\delta \beta^{-1} \log Z_q(p_t)}{\delta p_t} + \beta^{-1} \log p_t \right) \right). \tag{8}$$

Let $F_{p,\beta}(p) := \beta^{-1} \log Z_q(p) + \beta^{-1} S(p)$. By Theorem 2, the dynamics (8) of $p_t$ is the Wasserstein gradient descent flow for minimizing $F_{p,\beta}(p)$. By the Proposition 3 below, $F_{p,\beta}$ is strongly convex with respect to $p$. Therefore, it is possible to prove global convergence for the dynamics (8), and thus the convergence for the quasistatic Wasserstein gradient flow for the minimax problem (2).

**Proposition 3.** *For any probability distributions $p_1$, $p_2$ in $\mathcal{P}(\Omega)$, and any $\lambda \in [0, 1]$, we have*

$$F_{p,\beta}(\lambda p_1 + (1 - \lambda) p_2) < \lambda F_{p,\beta}(p_1) + (1 - \lambda) F_{p,\beta}(p_2).$$

In practice the partition function $\log Z_q(p_t)$ in (8) seems hard to approximate, especially when $\Omega$ is in high dimensional spaces. However, we show in the following proposition that the variation of the partition function with respect to $p_t$ can be written as a simple form involving $q_t$. This property will be used to derive a particle method in Section 5

**Proposition 4.** *For any $p \in \mathcal{P}(\Omega)$, we have*

$$\frac{\delta \beta^{-1} \log Z_q(p)}{\delta p} = U(\cdot, q[p]), \tag{9}$$

*where $q[p]$ is defined in (7). Therefore, the dynamics (8) is equivalent with*

$$\partial_t p_t = \nabla \cdot \left( p_t \nabla \left( U(x, q[p_t]) + \beta^{-1} \log p_t \right) \right). \tag{10}$$

## 4 CONVERGENCE ANALYSIS

In this section, we analyze the convergence of the quasistatic dynamics (7), (8). First, we make the following assumptions on $K$.

**Assumption 1.** *Assume $K \in C^\infty(\Omega \times \Omega)$, which means $K$ has continuous derivatives of any order (with respect to both $x$ and $y$).*

Since $\Omega$ is compact, assumption 1 implies boundedness and Lipschitz continuity of any derivatives of $K$.

Now, we state our main theorem, which shows the convergence of QSWGF to the Nash equilibrium.

**Theorem 5.** *(**main theorem**) Assume Assumption 1 holds for $K$. Then, starting from any initial $p_0, q_0 \in \mathcal{P}(\Omega)$, the dynamics (7), (8) has a unique solution $(p_t, q_t)_{t \geq 0}$, and the solution converges weakly to the unique Nash equilibrium of (2), $(p^*, q^*)$, which satisfies the fixed point problem (3).*

Theorem 5 guarantees convergence of the quasistatic Wasserstein gradient flow for any $\beta$, giving theoretical endorsement to the discretized algorithm that we will introduce in the next section. Note that the initialization $q_0$ in the theorem is not important, because we assume $q$ achieves equilibrium immediately after the initialization.

**Remark 2.** *The assumption on $K$'s smoothness can be made weaker. For example, during the proof, up to 4-th order derivatives of $K$ is enough to give sufficient regularity to the solution of the dynamics. We make the strong assumption partly to prevent tedious technical analysis so as to focus on the idea and insights.*

**Proof sketch** We provide some main steps and ideas of the proof of the main theorem in this section. The detailed proof is put in the appendix.

By the last section, since $q_t$ is always at equilibrium, we only need to considering a Wasserstein gradient descent flow for $F_{p,\beta}(p)$. Therefore, we can build our analysis based on the theories in (Mei et al., 2018) and (Jordan et al., 1998). However, compared with the analysis therein, our theory deals with a new energy term—$\beta^{-1} \log Z_q(p)$, which has not been studied by previous works. From now on, let $E_{p,\beta}(p) = \beta^{-1} \log Z_q(p)$, and $\Psi(\cdot, p) = \frac{\delta E_{p,\beta}(p)}{\delta p}$. By simple calculation we have

$$\Psi(x, p) = U(x, q[p]) = \frac{1}{Z_q(p)} \int_\Omega K(x, y) \exp\left( \int_\Omega \beta K(x, y) p(x) dx \right) dy. \tag{11}$$

First, we study the free energy $F_{p,\beta}(p)$, and show that it has a unique minimizer which satisfies a fixed point condition. This is the result of the convexity of $F_{p,\beta}$. We have the following lemma.

**Lemma 1.** *Assume Assumption 1 holds for $K$. Then, $F_{p,\beta}$ has a unique minimizer $p^*$ that satisfies*

$$F_{p,\beta}(p^*) = \inf_{p \in \mathcal{P}(\Omega)} F_{p,\beta}(p^*).$$

*Moreover, $p^*$ is the unique solution of the following fixed point problem,*

$$p^* = \frac{1}{Z} \exp\left(-\beta \Psi(x, p^*)\right), \tag{12}$$

*where $Z$ is the normalization factor.*

Next, we want to show that any trajectory given by dynamics (10) will converge to the unique minimizer of $F_{p,\beta}$. To achieve this, we first study the existence, uniqueness, and regularity of the solution to (8), i.e. the trajectory indeed exists and is well behaved. Related results are given by the following lemma.

**Lemma 2.** *Assume Assumption 1 holds for $K$. Then, starting from any initial $p_0 \in \mathcal{P}(\Omega)$, the weak solution $(p_t)_{t \geq 0}$ to (8) exists and is unique. Moreover, $(p_t)$ is smooth on $(0, \infty) \times \Omega$.*

The proof of Lemma 2 is based on Proposition 5.1 of (Jordan et al., 1998). Especially, the existence part is proven using the JKO scheme proposed in (Jordan et al., 1998). We consider a sequence of probability distributions given by the following discrete iteration schemes with time step $h$,

$$p_0^h = p_0, \quad p_k^h = \arg\min_{p \in \mathcal{P}(\Omega)} \left\{ \frac{1}{2} W_2^2(p, p_{k-1}^h) + h F_{p,\beta}(p) \right\}, \quad k > 0,$$

where $W_2(p, q)$ means the 2-Wasserstein distance between probability distributions $p$ and $q$. Let $(p_t^h)_{t \geq 0}$ be the piecewise constant interpolations of $(p_k^h)_{k \geq 0}$ on time. We show $(p_t^h)$ converges weakly (after taking a subsequence) to a weak solution of (8) as $h$ tends to 0. Details are given in the appendix.

Finally, noting that $F_{p,\beta}$ is a Lyapunov function of the dynamics (8), we have the following lemma showing the convergence of $(p_t)_{t \geq 0}$ to the solution of the Boltzmann fixed point problem (12). This finishes the proof of the main theorem.

**Lemma 3.** *Let $(p_t)_{t \geq 0}$ be the solution of (8) from any initial $p_0 \in \mathcal{P}(\Omega)$. Let $p^*$ be the unique minimizer of $F_{p,\beta}$ given by (12). Then, $p_t$ converges to $p^*$ weakly as $t \to \infty$.*

As a byproduct, since our convergence results does not impose requirement on $\beta$, if one is interested in the minimax problem without entropy regularization,

$$\min_{p \in \mathcal{P}(\Omega)} \max_{q \in \mathcal{P}(\Omega)} E(p, q), \tag{13}$$

then, Theorem 5 in (Domingo-Enrich et al., 2020) ensures that the quasistatic dynamics converges to approximate Nash equilibrium of (13) as long as $\beta^{-1}$ is small enough. Specifically, a pair of probability distributions $(p, q)$ is called $\epsilon$-Nash equilibrium of (13) if

$$\sup_{q' \in \mathcal{P}(\Omega)} E(p, q') - \inf_{p' \in \mathcal{P}(\Omega)} E(p', q) \leq \epsilon.$$

Then, we have the following theorem as a direct results of Theorem 5 in (Domingo-Enrich et al., 2020):

**Theorem 6.** *Let $C_K$ be the bound of $K$ that satisfies $|K(x, y)| \leq C_K$ for any $x, y \in \Omega$, and let $Lip(K)$ be the Lipschitz constant of $K$. For any $\epsilon > 0$, let $\delta = \epsilon/(2Lip(K))$, and let $V_\delta$ be the volume of a ball with radius $\delta$ in $\Omega$. Then, as long as*

$$\beta > \frac{4}{\epsilon} \log \left( \frac{2(1 - V_\delta)}{V_\delta} \left( \frac{4C_K}{\epsilon} - 1 \right) \right),$$

*there exists $T > 0$ which depends on $\epsilon$, such that for any $t > T$, the solution $p_t, q_t$ of the dynamics (7) and (8) at $t$ satisfies*

$$\sup_{q' \in \mathcal{P}(\Omega)} E(p_t, q') - \inf_{p' \in \mathcal{P}(\Omega)} E(p', q_t) \leq \epsilon.$$

## 5 THE QUASISTATIC LANGEVIN GRADIENT DESCENT-ASCENT METHOD

It is well known that PDEs with the form

$$\partial_t p(t, x) = \nabla \cdot (p(t, x)\mu(t, x)) + \lambda \Delta p(t, x)$$

are Fokker-Planck equations for SDEs $dX_t = -\mu(t, X_t)dt + \sqrt{2\lambda}dW_t$, and the solution for the PDE characterizes the law of $X_t$—the solution of the SDE—at any time. This result connects the Wasserstein gradient flow with SDE, and gives a natural particle discretization to approximate the continuous Wasserstein gradient flow. For example, the Wasserstein gradient descent-ascent flow dynamics (4) is the Fokker-Planck equation of the SDEs

$$dX_t = -\nabla_x U(X_t, q_t)dt + \sqrt{2\beta^{-1}}dW_t$$
$$dY_t = \nabla_y V(Y_t, p_t)dt + \sqrt{2\beta^{-1}}dW_t',$$

where $p_t$ and $q_t$ are the laws of $X_t$ and $Y_t$, respectively, and $W_t$ and $W_t'$ are two Brownian motions. Note that we have

$$\nabla_x U(x, q) = \int_\Omega \nabla_x K(x, y)q(y)dy, \quad \nabla_y V(y, p) = \int_\Omega \nabla_y K(x, y)p(x)dx.$$

Therefore, i.i.d. picking $X_0^{(i)} \sim p_0$ and $Y_0^{(i)} \sim q_0$ for $i = 1, 2, ..., n$, the particle update scheme, named Langevin Gradient Descent-Ascent (LGDA),

$$X_{k+1}^{(i)} = X_k^{(i)} - \frac{h}{n} \sum_{j=1}^n \nabla_x K(X_k^{(i)}, Y_k^{(j)}) + \sqrt{2h\beta^{-1}}\xi_k^{(i)},$$

$$Y_{k+1}^{(i)} = Y_k^{(i)} + \frac{h}{n} \sum_{j=1}^n \nabla_y K(X_k^{(j)}, Y_k^{(i)}) + \sqrt{2h\beta^{-1}}\zeta_k^{(i)}, \quad (14)$$

approximately solves the SDEs, and thus the empirical distributions of $X_k^{(i)}$ and $Y_k^{(i)}$ approximate the solutions of (4) when $n$ is large. Here, $\xi_k^{(i)}$ and $\zeta_k^{(i)}$ are i.i.d. samples from the standard Gaussian.

**Quasistatic Langevin gradient descent method**  Similarly, the dynamics (8) for $p$ is the Fokker-Planck equation for the SDE

$$dX_t = -\nabla \Psi(x, p_t)dt + \sqrt{2\beta^{-1}}dW_t, \quad (15)$$

where $p_t$ is the law of $X_t$. By proposition 4 we have $\Psi(x, p_t) = U(x, q[p_t])$. Hence, (15) can be written as

$$dX_t = -\nabla_x U(X_t, q[p_t])dt + \sqrt{2\beta^{-1}}dW_t, \quad (16)$$

with $q[p_t]$ at the equilibrium of the maximization problem (5), which can be attained by solving the SDE

$$dY_t = \nabla_y V(Y_t, p_t)dt + \sqrt{2\beta^{-1}}dW_t' \quad (17)$$

for sufficiently long time. This motivates us to design a quasistatic particle method as a discretization for the quasistatic Wasserstein gradient flow. Specifically, the method consists of an inner loop and an outer loop. The method starts from some particles $X_0^{(i)}$ and $Y_0^{(i)}$, $i = 1, 2, ..., n$, sampled i.i.d. from $p_0$ and $q_0$, respectively. Then, at the $k$-th step, the inner loop conducts enough iterations on the $Y$ particles to solve (17) with $p_t$ fixed (i.e. with the $X$ particles fixed), which drives the empirical distribution of $\{Y_k^{(i)}\}_{i=1}^n$ near equilibrium before each update of the outer loop. Next, the outer loop updates $X_k^{(i)}$ according the SDE (16). The algorithm is summarized in Algorithm 1.

---

**Algorithm 1:** Quasistatic Langevin gradient descent method (QSLGD)

---

**input** : $n_x, n_y, k_0, k_1, k_2, T \in \mathbb{N}_+, h_x, h_y > 0, p_0, q_0 \in \mathcal{P}(\Omega)$

**output:** Final particles $(X_T^{(i)}, Y_T^{(i)})_{i=1}^n$

---

1   Sample $(X_0^{(i)})_{i=1}^{n_x}$ i.i.d. from $p_0$, and $(Y_0^{(i)})_{i=1}^{n_y}$ i.i.d. from $q_0$;

2   $Y_{0,0}^{(i)} \leftarrow Y_0^{(i)}$ for $i = 1, 2, ..., n_y$;

3   **for** $s \leftarrow 1$ **to** $k_0$ **do**

4     $Y_{0,s}^{(i)} \leftarrow Y_{0,s-1}^{(i)} + \frac{h_y}{n_x} \sum_{j=1}^{n_x} \nabla_y K(X_0^{(j)}, Y_{0,s-1}^{(i)}) + \sqrt{2h_y\beta^{-1}}\xi$ ;       /\* **remark 3.1** \*/

5   **end**

6   $Y_0^{(i)} \leftarrow Y_{0,k_0}^{(i)}$ for $i = 1, 2, ..., n_y$;

7   **for** $t \leftarrow 1$ **to** $T$ **do**

8     $Y_{t-1,0}^{(i)} \leftarrow Y_{t-1}^{(i)}$ for $i = 1, 2, ..., n_y$;

9     **for** $s \leftarrow 1$ **to** $k_1$ **do**

10       $Y_{t-1,s}^{(i)} \leftarrow Y_{t-1,s-1}^{(i)} + \frac{h_y}{n_x} \sum_{j=1}^{n_x} \nabla_y K(X_{t-1}^{(j)}, Y_{t-1,s-1}^{(i)}) + \sqrt{2h_y\beta^{-1}}\xi$;

11     **end**

12     **for** $s \leftarrow 1$ **to** $k_2$ **do**

13       $Y_{t-1,s+k_1}^{(i)} \leftarrow Y_{t-1,s+k_1-1}^{(i)} + \frac{h_y}{n_x} \sum_{j=1}^{n_x} \nabla_y K(X_{t-1}^{(j)}, Y_{t-1,s+k_1-1}^{(i)}) + \sqrt{2h_y\beta^{-1}}\xi$;

14       $\hat{Y}_{t-1}^{((s-1)n_y+i)} \leftarrow Y_{t-1,s+k_1}^{(i)}$, for $i = 1, 2, ..., n_y$;

15     **end**

16     $X_t^{(i)} \leftarrow X_{t-1}^{(i)} + \frac{h_x}{k_2 n_y} \sum_{j=1}^{k_2 n_y} \nabla_y K(X_{t-1}^{(i)}, \hat{Y}_{t-1}^{(j)}) + \sqrt{2h_x\beta^{-1}}\xi$ ;       /\* **remark 3.2** \*/

17     $Y_t^{(i)} \leftarrow Y_{t-1,k_1+k_2}^{(i)}$, for $i = 1, 2, ..., n_y$;

18   **end**

---

**Remark 3.** *Generally speaking, Algorithm 1 consists of two nested loops. The inner loop solves $Y$ particles to equilibrium in each step of the outer loop, while the outer loop makes one iteration every time, using the equilibrium $Y$ particles. In the following are some additional explanation for the Algorithm:*

*1* **line 4:** *at the beginning of the algorithm, we conduct $k_0$ additional inner iterations for $Y$, where $k_0$ may be a large number. This is because at the beginning the $Y$ particles are far from equilibrium. In later outer iterations, since each time the $X$ particles only move for a small distance, the $Y$ particles are close to the equilibrium. Therefore, $k_1$ and $k_2$ need not to be large.*

*2* **line 17:** *In each inner loop, we conduct $k_1 + k_2$ inner iterations for the $Y$ particles, and collect those from the last $k_2$ iterations. We use these $k_2 n$ particles in the update of $X$ particles to approximate the distribution $q[p]$. We assume during the last $k_2$ inner iterations the $Y$ particles are at equilibrium. One can take $k_2$ to be 1 if $n_y$ is large enough, while taking large $k_2$ allows smaller number of $Y$ particles.*

## 5.1 EXAMPLES

In this section, we apply the quasistatic Langevin gradient descent method to several problems.

**1-dimensional game on torus**   We first consider a problem with $x$ and $y$ on the 1-dimensional torus. Specifically, we consider

$$K(x, y) = \sin(2\pi x)\sin(2\pi y),$$

where $x, y \in \mathbb{R}/\mathbb{Z}$. It is easy to show that, with this $K$ and a positive $\beta$, at the Nash equilibrium of the problem (1) $p$ and $q$ are both uniform distributions. We take initial distributions $p_0$ and $q_0$

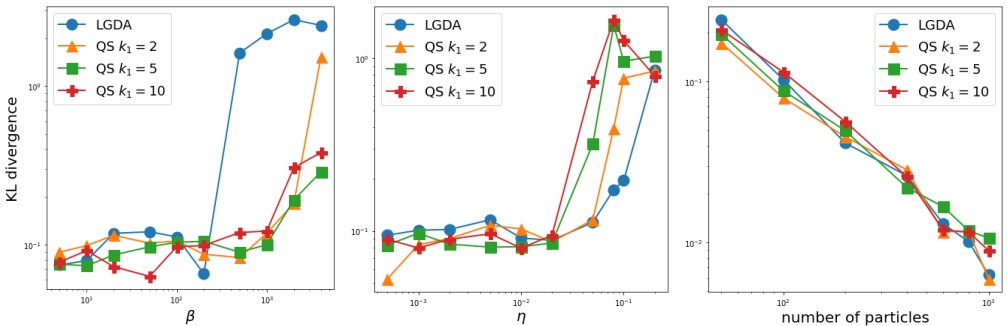

Figure 1: Experiment results with $K(x, y) = \sin(2\pi x)\sin(2\pi y)$. The three figures show the KL divergence of the empirical particle distribution to the uniform distribution of LGDA and QSLGD at different $\beta$, $\eta$ and number of particles. Each point is an average of 5 experiments.

to be the uniform distribution on $[0, 1/4]$. Figure 1 shows the comparison of the quasistatic particle method with LGDA for different $\beta$, step length, and number of particles. In the experiments, all quasistatic methods take $k_0 = 1000$ and $k_2 = 1$, with different $k_1$ shown in the legends. For each experiment, we conduct 300000, 150000, 60000, 30000 outer iterations for LGDA, QS2, QS5, and QS10, respectively. We take different different numbers of iterations for different methods in the consideration of different number of inner iterations. The error is then computed after the last iteration, measured by the KL divergence of the empirical distribution given by particles and the uniform distribution (both in the forms of histograms with 10 equi-length bins). Each point in the figures is an average of 5 experiments.

Seen from the left figure, the QSLGD has comparable performance than LGDA when $\beta$ is small, in which case diffusion dominates the dynamics, while it performs much better than LGDA when $\beta$ is large. We can also see better tolerance to large $\beta$ when more inner iterations are conducted. This shows the advantage of the QSLGD over LGDA when the regularization stength is weak. The middle figures shows slightly better performance of the QSLGD when the step length $\eta$ (both $\eta_x$ and $\eta_y$) is small. However, when $\eta$ is big, LGDA tends to give smaller error. The results may be caused by the instability of the inner loop when $\eta$ is big. It also guides us to pick small step length when applying the proposed method. Finally, the right figure compares the influence of the number of particles when $\beta = 100$ and $\eta = 0.01$, in which case the two methods perform similarly. We can see that the errors for both methods scale in a $1/n$ rate as the number of particles $n$ changes.

**Polynomial games on spheres** In the second example, we consider a polynomial games on sphere similar to that studied in (Domingo-Enrich et al., 2020),

$$K(x, y) = x^T A_0 x + x^T A_1 y + y^T A_2 y + y^T A_3 (x^2), \tag{18}$$

where $x, y \in \mathbb{S}^{d-1}$ and $(x^2)$ is the element-wise square of $x$. In this problem, we consider the Nash equilibrium of $\min_p \max_q E(p, q)$. Hence, we take big $\beta$ (small $\beta^{-1}$) and compare the Nikaido and Isoda (NI) error of the solutions found by different methods (Nikaidô & Isoda, 1955). The NI error is defined by

$$NI(p, q) := \sup_{q' \in \mathcal{P}(\Omega)} E(p_t, q') - \inf_{p' \in \mathcal{P}(\Omega)} E(p', q_t),$$

which is also used in Theorem 6. The left panel of Figure 2 shows the NI errors of the solutions found by different methods with different dimensions, we see comparable performance of the QSLGD with LGDA.

**GANs** Finally, we test our methods on the training of GANs. We train GANs to learn Gaussian mixtures. The results after training are shown in the middle and right panels of Figure 2, where Gaussian mixtures with 4 and 8 modes are learned, respectively. We train GANs with 5 generators and 5 discriminators, and take $k_0 = 100, k_1 = 5, k_2 = 1$. The results show that the mixture of GANs trained by QSLGD can learn Gaussian mixtures successfully.

In the right panel of Figure 2, we show the results of learning high dimensional Gaussian mixtures. In the d-dimensional experiment, the Gaussian mixture has $d$ modes centered at $\mathbf{e}_1, \mathbf{e}_2, ..., \mathbf{e}_d$ with

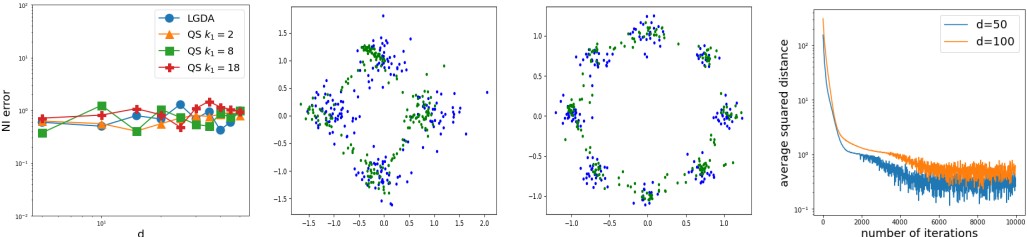

Figure 2: **(Left)** The NI error of the solutions found by different algorithms for the polynomial game (18), at different dimensions. Each point is an average of 10 experiments. **(Middle left, Middle right)** Generation results of mixture of GANs. The blue points are sampled from groundtruth distribution, while the green points are generated by the a mixture of generators. **(Right)** The average squared distance of generated data to closest mode center for learning high dimensional Gaussian mixtures.

standard deviation 0.1. Here, $\mathbf{e}_i$ is the $i$-th unit vector in the standard basis of $\mathbb{R}^d$. Model and algorithm with same hyper-parameters as above are used. In the figure, we measure the average squared distance of the generated data to the closest mode center along the training process. The figure shows that the average squared distance can be reduced to $0.3 - 0.5$ after 10000 iterations. While the ideal value is 0.1, the current results still show that the learnt distribution concentrates at the mode centers. Better results may be obtained after longer training or careful hyper-parameter tuning.

## 6 DISCUSSION

In this paper, we study the quasistatic Wasserstein gradient flow for the mixed Nash equilibrium problem. We theoretically show the convergence of the continuous dynamics to the unique Nash equilibrium. Then, a quasistatic particle method is proposed by discretizing the continuous dynamics. The particle method consists of two nested loops, and conduct sufficient inner loop in each step of the outer loop. Numerical experiments show the effectiveness of the method. Comparison with LGDA shows the proposed method has advantage over LGDA when $\beta$ is large (which is usually the case of interest), and performs as good as LGDA in most other cases.

Theoretical extensions are possible. For example, strong convergence results may be established by similar approaches taken in (Feng & Li, 2020). We leave this as future work.

In practice, the idea of nested loops is not new for minimax optimization problems. It is already discussed and utilized in the earliest works for GANs (Goodfellow et al., 2014a), and Wasserstein GANs (Arjovsky et al., 2017). In those works, the discriminator is updated for several steps each time the generator is updated. Our work is different from these works because we consider mixed Nash equilibrium and hence our method is particle based, while their method searches for pure Nash equilibrium.

Finally, though particle methods finding mixed Nash equilibria have stronger theoretical guarantees, applying these methods to the training of GANs faces the problem of computational cost. With both the generator and discriminator being large neural networks, training mixture of GANs with many generators and discriminators imposes formidable computational cost. Developing more efficient particle methods for GANs is an important future work.

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

# A    PROOFS FOR SECTION 3

## A.1    PROOF OF THEOREM 2

Note that the free energy in (5) can be written as

$$\int_\Omega V(y, p_t)q(y)dy - \beta^{-1}S(q), \tag{19}$$

in which the first term is linear with respect to $q$. Hence, a calculation with Lagrange multiplier shows (19) has a unique minimizer $q[p_t]$ with the form of a Gibbs distribution (e.g. see Chapter 4 of (Mezard & Montanari, 2009)):

$$q[p_t](y) = \frac{1}{Z_q(p_t)}\exp(\beta V(y, p_t)). \tag{20}$$

Next, we consider the free energy for $p_t$ when $q$ is at the equilibrium. By (19) we have

$$E(p_t, q[p_t]) = \int_\Omega V(y, p_t)q[p_t](y)dy$$
$$= \frac{1}{Z_q(p_t)}\int_\Omega V(y, p_t)\exp(\beta V(y, p_t))dy. \tag{21}$$

On the other hand, we have

$$\beta^{-1}S(q[p_t]) = \beta^{-1}\int_\Omega \frac{1}{Z_q(p_t)}\exp(\beta V(y, p_t))\log\left(\frac{1}{Z_q(p_t)}\exp(\beta V(y, p_t))\right)dy$$
$$= \beta^{-1}\int_\Omega \frac{1}{Z_q(p_t)}\exp(\beta V(y, p_t))\left(\beta V(y, p_t) - \log Z_q(p_t)\right)dy$$
$$= \frac{1}{Z_q(p_t)}\int_\Omega V(y, p_t)\exp(\beta V(y, p_t))dy - \beta^{-1}\log Z_q(p_t). \tag{22}$$

Combining (21) and (22), we obtain

$$E(p_t, q[p_t]) + \beta^{-1}S(p_t) - \beta^{-1}S(q[p_t]) = \beta^{-1}\log Z_q(p_t) + \beta^{-1}S(p_t).$$

Therefore, the dynamics of $p_t$ is the Wasserstein gradient descent flow minimizing the free energy $\beta^{-1}\log Z_q(p_t) + \beta^{-1}S(p_t)$, given by

$$\partial_t p_t = \nabla \cdot \left(p_t \nabla \left(\frac{\delta\beta^{-1}\log Z_q(p_t)}{\delta p_t} + \beta^{-1}\log p_t\right)\right).$$

This finishes the proof.

## A.2    PROOF OF PROPOSITION 3

Since $S(p)$ is strongly convex, it suffices to show $\log Z_q(p)$ is convex. Recall that

$$\log Z_q(p) = \log\left(\int_\Omega \exp(\beta V(y, p))dy\right).$$

Note that $V(\cdot, p) = \int K(x, \cdot)p(x)dx$ is linear with respect to $p$, we have

$$V(\cdot, \lambda p_1 + (1 - \lambda)p_2) = \lambda V(\cdot, p_1) + (1 - \lambda)V(\cdot, p_2).$$

Hence,

$$\log Z_q(\lambda p_1 + (1 - \lambda)p_2) = \log\left(\int_\Omega \exp(\beta V(y, \lambda p_1 + (1 - \lambda)p_2))dy\right)$$
$$= \log\left(\int_\Omega \exp(\beta\lambda V(y, p_1)) \cdot \exp(\beta(1 - \lambda)V(y, p_2))\right)$$
$$\leq \log\left(\left(\int_\Omega \exp(\beta V(y, p_1))\right)^\lambda \left(\int_\Omega \exp(\beta V(y, p_2))\right)^{1-\lambda}\right)$$
$$= \lambda \log Z_q(p_1) + (1 - \lambda)\log Z_q(p_2). \tag{23}$$

The second last line is given by the Hölder inequality.

### A.3 PROOF OF PROPOSITION 4

The proposition follows from the following derivations.

$$
\begin{aligned}
\frac{\delta \beta^{-1} \log Z_q(p)}{\delta p} &= \beta^{-1} \frac{1}{Z_q(p)} \frac{\delta Z_q(p)}{\delta p} \\
&= \beta^{-1} \frac{1}{Z_q(p)} \int_\Omega \exp(\beta V(y,p)) \beta K(x,y) dy \\
&= \int_\Omega K(x,y) \frac{\exp(\beta V(y,p))}{Z_q(p)} dy \\
&= \int_\Omega K(x,y) q[p](y) dy \\
&= U(x, q[p]).
\end{aligned}
\tag{24}
$$

## B PROOF OF THEOREM 5

In this section, we prove our main theorem. The proof will follow the sketch given in Section 4. Given the assumptions on $K$ and the conclusions of Lemma 1 and Lemma 2, Lemma 3 is a direct result of Lemma 10.12 in (Mei et al., 2018), which we will ignore the proof. In the following, we show Lemma 1 and Lemma 2. Some techniques in the proof come from (Jordan et al., 1998) and (Mei et al., 2018).

### B.1 PROOF OF LEMMA 1

Our proof follows the proof of Proposition 4.1 in (Jordan et al., 1998). First, we show the existence of the minimizer for $F_{p,\beta}$. To see this, note that $K$ is bounded on $\Omega$. Assume $C_K$ is a constant such that $|K(x,y)| \leq C_K$ for any $x, y \in \Omega$. Then, we have

$$
E_{p,\beta}(p) = \int_\Omega U(x, q[p]) p(x) dx = \int_{\Omega \times \Omega} K(x,y) q[p](y) p(x) dx dy \geq -C_K,
$$

and

$$
S(p) = \int_\Omega p(x) \log p(x) dx \geq \int_\Omega -\frac{1}{e} dx = -\frac{1}{e}.
$$

This means $F_{p,\beta}(p)$ is lower bounded, i.e. $\inf_p F_{p,\beta}(p) > -\infty$. Hence, we can find a sequence $(p_k)_{k=1}^\infty$ such that

$$
\lim_{k \to \infty} F_{p,\beta}(p_k) = \inf_p F_{p,\beta}(p).
$$

Similar to (Jordan et al., 1998), we can show boundedness of $\{\int \max\{p_k \log p_k, 0\} dx\}$ and $\{\int p_k^2 dx\}$, which implies that $(p_k)$ is uniformly integrable, and thus there exists a weakly convergent subsequence of $(p_k)$.

Without loss of generality, assume $p_k \rightharpoonup p^*$ in $L^1(\Omega)$. Then we need to show $p^*$ is a minimizer of $F_{p,\beta}$. By (Jordan et al., 1998), the entropy term satisfies

$$
S(p^*) \leq \liminf_{k \to \infty} S(p_k).
$$

Hence, the conclusion follows if $E_{p,\beta}$ is continuous in the weak topology. To show this, first note that for any $p \in \mathcal{P}(\Omega)$, we have

$$
\int_\Omega \exp(\beta V(y,p)) dy \geq e^{-\beta C_K}.
$$

Because the function $\log(x)$ is $1/c$-Lipschitz for $x \in [c, \infty]$, for any $p_k$ we have

$$
\left| \beta^{-1} \log Z_q(p_k) - \beta^{-1} \log Z_q(p^*) \right| \leq \beta^{-1} e^{\beta C_K} \left| \int_\Omega \left( e^{\beta V(y, p_k)} - e^{\beta V(y, p^*)} \right) dy \right|.
$$

By similar boundedness argument, we have

$$\left| \int_\Omega \left( e^{\beta V(y,p_k)} - e^{\beta V(y,p^*)} \right) dy \right| \le \int_\Omega \left| e^{\beta V(y,p_k)} - e^{\beta V(y,p^*)} \right| dy$$

$$\le e^{\beta C_K} \int_\Omega \beta \left| V(y,p_k) - V(y,p^*) \right| dy$$

$$\le \beta e^{\beta C_K} \int_\Omega \left| \int_\Omega K(x,y)(p_k(x) - p^*(x))dx \right| dy$$

Totally we have

$$\left| \beta^{-1} \log Z_q(p_k) - \beta^{-1} \log Z_q(p^*) \right| \le e^{2\beta C_K} \int_\Omega \left| \int_\Omega K(x,y)(p_k(x) - p^*(x))dx \right| dy.$$

Since $K$ is bounded and Lipschitz, it is easy to show that

$$\lim_{k \to \infty} \int_\Omega \left| \int_\Omega K(x,y)(p_k(x) - p^*(x))dx \right| dy = 0.$$

Therefore, we have

$$F_{p,\beta}(p^*) \le \liminf_{k \to \infty} F_{p,\beta}(p_k) = \inf_p F_{p,\beta}(p),$$

and thus $F_{p,\beta}(p^*) = \inf_p F_{p,\beta}(p)$.

Next, we show $p^*$ satisfies the fixed point condition

$$p^* = \frac{1}{Z} \exp\left( -\beta \Psi(x,p^*) \right). \tag{25}$$

This follows the proof of Lemma 10.3 in (Mei et al., 2018), by first showing $p^*$ has full support on $\Omega$, and then showing

$$\Psi(x,p^*) + \beta^{-1} \log p^*(x)$$

is a constant.

Finally, we show $p^*$ is unique following Lemma 10.4 of (Mei et al., 2018). Specifically, we show the Boltzmann fixed point problem (25) only has one solution by the convexity of $F_{p,\beta}$. Assume (25) has two different solutions $p_1$ and $p_2$, i.e.

$$p_1 = \frac{1}{Z(p_1)} \exp\left( -\beta \Psi(x,p_1) \right), \ \ p_2 = \frac{1}{Z(p_2)} \exp\left( -\beta \Psi(x,p_2) \right).$$

Then, we have

$$\log Z(p_1) = -\beta \Psi(x,p_1) - \log p_1(x),$$
$$\log Z(p_2) = -\beta \Psi(x,p_2) - \log p_2(x).$$

Taking difference of the above two equations, and integrating with $p_1 - p_2$, we have

$$0 = \int_\Omega (\log Z(p_1) - \log Z(p_2))(p_1(x) - p_2(x))dx$$

$$= -\beta \int_\Omega (\Psi(x,p_1) - \Psi(x,p_2))(p_1(x) - p_2(x))dx - \int_\Omega (\log p_1(x) - \log p_2(x))(p_1(x) - p_2(x))dx. \tag{26}$$

By the monotonicity of $\log x$, the second term of (26) is non-negative, and takes zero only if $p_1 = p_2$. For the first term, recall that we have proven in Proposition 3 that $E_{p,\beta}$ is convex, we then have

$$E_{p,\beta}(p_1) \ge E_{p,\beta}(p_2) + \int_\Omega \Psi(x,p_2)(p_1(x) - p_2(x))dx,$$

and

$$E_{p,\beta}(p_2) \ge E_{p,\beta}(p_1) + \int_\Omega \Psi(x,p_1)(p_2(x) - p_1(x))dx.$$

Taking difference of the two equations gives

$$\int_\Omega (\Psi(x,p_1) - \Psi(x,p_2))(p_1(x) - p_2(x))dx \ge 0.$$

Therefore, (26) holds if and only if $p_1 = p_2$. This finishes the proof of uniqueness of $p^*$, and also completes the proof of Lemma 1.

## B.2 PROOF OF LEMMA 2

We show the existence of the weak solution using the JKO scheme used in (Jordan et al., 1998). Let $\text{dist}(\cdot, \cdot)$ be the distance metric on $\Omega$. Then, the 2-Wasserstein distance is on $\mathcal{P}(\Omega)$ is defined as

$$
W_2(p_1, p_2) := \left( \inf_{\gamma \in \Gamma(p_1, p_2)} \int_{\Omega \times \Omega} \text{dist}(x_1, x_2)^2 d\gamma(x_1, x_2) \right)^{1/2},
$$

where $\Gamma(p_1, p_2)$ contains all couplings of $p_1$ and $p_2$, i.e. probability distributions on $\Omega \times \Omega$ with first and second marginals being $p_1$ and $p_2$, respectively. Then, for any $h > 0$, we consider the sequence of probability distributions $(p_k^h)_{k=0}^{\infty}$ obtained by the following iteration scheme:

$$
p_0^h = p_0, \quad p_k^h = \arg \min_{p \in \mathcal{P}(\Omega)} \left\{ \frac{1}{2} W_2^2(p, p_{k-1}^h) + h F_{p,\beta}(p) \right\}, \quad k > 0, \tag{27}
$$

By similar argument of Proposition 4.1 in (Jordan et al., 1998), each minimization problem in (27) has a unique solution. Hence, $(p_k^h)_{k=0}^{\infty}$ is uniquely defined. Let $p_t^h$ be the piecewise constant interpolation of $(p_k^h)$ on $t$, i.e.

$$
p_t^h = p_k^h, \text{ for } t \in [kh, (k+1)h),
$$

for $k = 0, 1, 2, \dots$. We now show that there exists a subsequence of $h_n \to 0$ and a $p_t$ such that $p_t^{h_n} \rightharpoonup p_t$ on $(0, T) \times \Omega$ for any $T > 0$ and $p_t$ is a weak solution of (8). This is proven in two steps:

1. The existence of weakly convergence subsequence, and

2. $p_t^h$ approximately satisfies the equation (8).

For the first point, we show uniform integrability by showing

$$
\int_{\Omega} \|x\|^2 p_k^h(x) dx \leq C \tag{28}
$$

and

$$
\int_{\Omega} \max\{p_k^h \log p_k^h, 0\} dx \leq C \tag{29}
$$

for any $h$ and $k \geq 0$, and an absolute constant $C$. Equation (28) follows directly from the compactness of $\Omega$. For (29), note that for any $h$ and $k \geq 0$ we have

$$
\frac{1}{2} W_2^2(p_k^h, p_{k-1}^h) + h F_{p,\beta}(p_k^h) \leq h F_{p,\beta}(p_{k-1}^h),
$$

which implies

$$
F_{p,\beta}(p_k^h) \leq F_{p,\beta}(p_{k-1}^h).
$$

Therefore,

$$
\begin{aligned}
\int_{\Omega} \max\{p_k^h \log p_k^h, 0\} dx &\leq S(p_k^h) + \int_{\Omega} \left| \min\{p_k^h \log p_k^h, 0\} \right| dx \\
&\leq S(p_k^h) + \int_{\Omega} \frac{1}{e} dx \\
&\leq F_{p,\beta}(p_k^h) - E_{p,\beta}(p_k^h) + \int_{\Omega} \frac{1}{e} dx \\
&\leq F_{p,\beta}(p_0^h) - E_{p,\beta}(p_k^h) + \int_{\Omega} \frac{1}{e} dx.
\end{aligned}
$$

Since $K$ is bounded, $E_{p,\beta}$ is bounded, and thus the above expression is also bounded, which gives (29). With (28) and (29), there exists $p_t(x)$ and a sequence $(h_n)$ with $h_n \to 0$, such that $p_t^{h_n} \rightharpoonup p_t$ in $L^1((0, T) \times \Omega)$ for any $T > 0$. Moreover, $p_t \in \mathcal{P}(\Omega)$ for almost every $T$. By changing $p_t$ on a zero measure set of $t$, we can assume $p_t \in \mathcal{P}(\Omega)$ for any $t \in (0, \infty)$. With the same analysis as (Jordan et al., 1998), the weak convergence can happen for any $t$, i.e. $p_t^{h_n} \rightharpoonup p_t$ in $L^1(\Omega)$ for any $t \in (0, \infty)$.

For the second point, similar to (Jordan et al., 1998), consider any vector field $\xi \in C^\infty(\Omega, \Omega)$ and the corresponding flux $\Phi_\tau$ given by

$$\partial_\tau \Phi_\tau = \xi(\Phi_\tau), \quad \Phi_0(x) = x,$$

and let $q_\tau = \Phi_\tau \sharp p_k^h$, then we have

$$\frac{1}{\tau} \left( \left( \frac{1}{2} W_2^2(p_{k-1}^h, q_\tau) + h F_{p,\beta}(q_\tau) \right) - \left( \frac{1}{2} W_2^2(p_{k-1}^h, p_k^h) + h F_{p,\beta}(p_k^h) \right) \right) \geq 0 \qquad (30)$$

for any $\tau > 0$. We need to study the limit when $\tau \to 0^+$. By the calculation in (Jordan et al., 1998) we have

$$\limsup_{\tau \to 0^+} \frac{1}{\tau} \left( \frac{1}{2} W_2^2(p_{k-1}^h, q_\tau) - \frac{1}{2} W_2^2(p_{k-1}^h, p_k^2) \right) \leq \int_{\Omega \times \Omega} (y - x)\xi(y) d\gamma(x,y), \qquad (31)$$

and

$$\left. \frac{d}{d\tau} S(q_\tau) \right|_{\tau=0} = -\int_\Omega p_k^h \nabla \cdot \xi dx, \qquad (32)$$

where the $\gamma$ in (31) is the optimal transport between $p_{k-1}^h$ and $p_k^h$. For the $E_{p,\beta}$ term, we have

$$
\begin{aligned}
\lim_{\tau \to 0^+} \frac{1}{\tau} \left( E_{p,\beta}(q_\tau) - E_{p,\beta}(p_k^h) \right) &= \lim_{\tau \to 0^+} \frac{1}{\beta \tau} \log \left[ \frac{\int_\Omega \exp(\beta V(y, q_\tau)) dy}{\int_\Omega \exp(\beta V(y, p_k^h)) dy} \right] \\
&= \lim_{\tau \to 0^+} \frac{1}{\beta \tau} \left[ \frac{\int_\Omega \exp(\beta V(y, q_\tau)) dy}{\int_\Omega \exp(\beta V(y, p_k^h)) dy} - 1 \right] \\
&= \frac{1}{\beta Z_q(p_k^h)} \lim_{\tau \to 0^+} \frac{1}{\tau} \left( \int_\Omega (\exp(\beta V(y, q_\tau)) - \exp(\beta V(y, p_k^h))) dy \right) \\
&= \frac{1}{\beta Z_q(p_k^h)} \int_\Omega \exp(\beta V(y, p_k^h)) \beta \left( \int_\Omega \nabla_x K(x,y) \cdot \xi(x) p_k^h(x) dx \right) dy \\
&= \int_\Omega \nabla_x \Psi(x, p_k^h) \cdot \xi(x) p_k^h(x) dx. \qquad (33)
\end{aligned}
$$

Combining the above result with (31) and (32), taking both $\xi$ and $-\xi$, we get from (30) that

$$\int_{\Omega \times \Omega} (y - x)\xi(y) d\gamma(x,y) + h \int_\Omega \nabla_x \Psi(x, p_k^h) \cdot \xi(x) p_k^h(x) dx - \frac{h}{\beta} \int_\Omega p_k^h \nabla \cdot \xi dx = 0 \qquad (34)$$

for any $\xi \in C^\infty(\Omega, \Omega)$. Then, following the derivation in (Jordan et al., 1998) (proof of Proposition 5.1), as well as the following control

$$\sum_{k=1}^N W_2^2(p_{k-1}^h, p_k^h) \leq Ch$$

for any $N$ that satisfies $Nh \leq T$ for some fixed $T$, we can integrate (34) over $t$ by viewing $p_k^h$ as $p_t^h$ at appropriate $t$, and take the limit $h_n \to 0$ and show that $p_t$ is a weak solution. During the limit, we need to pay special attention to the second term, i.e. the following limit, which is not dealt with in the reference:

$$\lim_{n \to \infty} \int_0^T \int_\Omega \nabla_x \Psi(x, p_t^{h_n}) \cdot \xi(x) p_t^{h_n}(x) dx = \int_0^T \int_\Omega \nabla_x \Psi(x, p_t) \cdot \xi(x) p_t(x) dx. \qquad (35)$$

We prove this by showing $\nabla_x \Psi(x, p_t^{h_n})$ converges to $\nabla_x \Psi(x, p_t)$ uniformly.

For any fixed $t$, recall that we have

$$p_t^{h_n} \rightharpoonup p_t.$$

Therefore, for any $y \in \Omega$, we have

$$\int_\Omega K(x,y) p_t^{h_n}(x) dx \to \int_\Omega K(x,y) p_t(x) dx.$$

Note that $\int_\Omega K(x, y)p(x)dx$ is uniformly continuous with respect to $y$ for any $p \in \mathcal{P}(\Omega)$, we can conclude that $\int_\Omega K(x, y)p_t^{h_n}(x)dx$ converges to $\int_\Omega K(x, y)p_t(x)dx$ uniformly over $y$. Hence, we have $Z_q(p_t^{h_n}) \to Z_q(p_t)$, and $q[p_t^{h_n}](y) \to q[p_t](y)$ uniformly. This further implies that

$$\nabla_x \Psi(x, p_t^{h_n}) = \int_\Omega \nabla_x K(x, y)q[p_t^{h_n}](y)dy$$

converges to $\nabla_x \Psi(x, p_t)$ for all $x \in \Omega$. This finishes the proof of (35), and also completes the proof that $p_t$ is a weak solution of of equation (8).

Now we have proven the existence of the weak solution. The regularity and uniqueness of the solution follows the same analysis of Proposition 5.1 in (Jordan et al., 1998).

