# OpenReview forum: "Provably convergent quasistatic dynamics for mean-field two-player zero-sum games"
_ICLR.cc/2022/Conference — ICLR 2022 Poster_

### Official Review · Reviewer_TF7Q · 2021-10-31

**Correctness:** 4
**Technical Novelty And Significance:** 3
**Empirical Novelty And Significance:** 3
**Recommendation:** 6
**Confidence:** 3

**Main Review:**

The paper is well-written and have clear objectives. It might be a good idea to add more experiments illustrating the power of the authors' method.

**Summary Of The Paper:**

This paper studies a minimax problem arising from finding the mixed Nash equilibrium for mean-field two player games. A quasi-static Wasserstein gradient flow is proposed to solve such optimization problems. By discretizing this flow, a simple while practical Langevin gradient descent algorithm is proposed.

**Summary Of The Review:**

Good paper. Clear writing.

---

> ### Author Response · Authors · 2021-11-22
> **Response to reviewer TF7Q**
>
> **It might be a good idea to add more experiments illustrating the power of the authors' method.**
>
> We thank the reviewer for the appreciation to our work. In the revised version of the paper, we added a numerical experiment in which our quasistatic Langevin gradient descent method is used to train mixed GANs to learn high-dimensional Gaussian mixture distributions (50 and 100 dimensions). This shows the effectiveness of our method on high-dimensional problems. Please see the revised paper, as well as the answer to question (2) of reviewer izUn for more details.

---

### Official Review · Reviewer_6QW7 · 2021-11-01

**Correctness:** 4
**Technical Novelty And Significance:** 3
**Empirical Novelty And Significance:** 3
**Recommendation:** 6
**Confidence:** 5

**Main Review:**

The paper is overall well written with clear mathematics.

Cons: The authors present the gradient flow formulation for zero-sum games. Here they complete the sup step and only consider the inf steps.

Weakness:

1. For the functional  F(p)=max_{q} E(p, q)-beta S(q). Did the author provide some geodesic convexity property of F in Wasserstein space?

2. Some motivations on Energy functional (1) are needed. The method seems to work for a general type of energy or loss function.

3. The authors miss important literature on Wasserstein dynamics in GANs, and mean-field games. I suggest the authors carefully compare them with the results in this paper.

Arbel, Gretton, Li, Montufar, Kernelized Wasserstein natural gradient, ICLR, 2019

Lin, Li, Montufar, Wasserstein proximal of GANs, GSI 2021.

Lin, et.al. Alternating the Population and Control Neural Networks to Solve High-Dimensional Stochastic Mean-Field Games,  arXiv:2002.10113.

Wang, Li, Accelerated information gradient flow, arXiv:1909.02102.

**Summary Of The Paper:**

In this paper, the authors study the minimax problem for finding the mixed Nash equilibrium for mean-field two-player zero-sum games. This is an inf-sup problem in the Wasserstein space. In this paper, the authors consider the case where the sup problem is solved and apply the gradient descent flow for the result inf sup functional. Some theoretical analyses are conducted on these dynamics. They show its convergence to the mixed Nash equilibrium under some conditions. Some numerical schemes for the quasistatic Langevin gradient descent method with inner-outer iterations are presented. They show that the method works on training the mixture of GANs.

**Summary Of The Review:**

The authors study a gradient flow of a particular type functional, which comes from an inf-sup problem in zero-sum games. The paper is overall well written. Some key issues on geodesic convexity and relation to literature are missed.

---

> ### Author Response · Authors · 2021-11-22
> **Response to reviewer 6QW7**
>
> We thank the reviewer for the comments. The questions raised by the reviewer are answered as follows.
>
> **1. For the functional F(p), did the author provide some geodesic convexity property of F in Wasserstein space?**
>
> The geodesic convexity for the energy of $p$ in the quasistatic Wasserstein flow ($F_{p,\beta}$ in the paper) is not hard to show. By definition we have
> $$F_{p,\beta} = \max_q (E(p,q)-\beta^{-1}S(q)) + \beta^{-1}S(p).$$
> For any fixed $q$, $E(p,q)-\beta^{-1}S(q)$ is linear with $p$, and thus it is geodesic convex. Since the maximum of convex functions is still convex, we have
> $\max_q (E(p,q)-\beta^{-1}S(q))$ is geodesic convex. Finally, considering $S(p)$ is geodesic convex, we finish showing the geodesic convexity of $F$.
>
> The geodesic convexity of the energy is not central to our theory. In the revised paper, we add some discussion to it after proposition 3.
>
> **2. Some motivations on Energy functional (1) are needed. The method seems to work for a general type of energy or loss function.**
>
> The energy functional (1) comes from the mixed Nash equilibrium problem (especially originating from the training of GANs). In the first term, the function $K$ is the payoff function of the two-player game. When finding mixed Nash equilibrium, we consider two probability distributions of strategies instead of two pure strategy. Hence, we consider a min-max problem of probability distributions $p$ and $q$. The total payoff is computed by the expectation of the payoff function when the strategies of the two players are sampled from $p$ and $q$, respectively. This gives the first term. The other two terms are entropy regularizations for $p$ and $q$. They are added to prevent the two distributions from being singular. The entropy regularizations help theoretical analysis. They also contribute the noise terms in the quasistatic Langevin gradient descent method, which enables exploration of the particles. Usually the regularization is not strong, so the Nash equilibrium of the regularized problem ((1) in the paper) is not far from the unregularized problem. This is shown by Theorem 6 in the paper, where we cite the analysis in [Domingo-Enrich et al.].
>
> **3. The authors miss important literature on Wasserstein dynamics in GANs, and mean-field games. I suggest the authors carefully compare them with the results in this paper.**
>
> We thank the reviewer for providing the related literature. We made a careful comparison of our work with the mentioned works in the revised related work section. The added discussion is as follows:
>
> *The Wasserstein gradient flow in the density space has been explored in previous works. For example, [Wang et al.] studied the Nesterov’s accelerated gradient flows for probability distributions under the Wasserstein metric, and [Arbel et al.] studied practical implementations of the natural gradient method for the Wasserstein metric. Both works focus on minimization problems instead of min-max problems considered in this work. A more related work is [Lin et al. 2021b], where a natural gradient based algorithm is proposed for training GANs. Yet, the method still optimizes one generator and one discriminator, searching for pure Nash equilibrium. Another work that derives algorithms for GANs from a Wasserstein perspective is [Lin et al.2021a].*

---

> > ### Comment · Reviewer_6QW7 · 2021-11-24
> > **Question 1**
> >
> > Thanks for your revision. I will keep my score and may increase it after addressing question 1.
> >
> > 1. For question 1, I am asking about the displacement convexity in Wasserstein-2 space. It is not the convexity in the L^2 space of density.
> > In this case, one is required to check the convexity along the Wasserstein geodesic. Here the linear function of p could lead to non-convexity in Wasserstein-2 space. This could also take effect in the convergence rate of the proposed flow.
> >
> > 2, 3 are great. The motivation and literature reviews address my questions.
> >
> > Overall, I suggest the publication of this paper.

---

> > > ### Author Response · Authors · 2021-11-26
> > > **May not be convex in Wasserstein space**
> > >
> > > We apologize for the misunderstanding we made in our first response. The functional is convex in L2 space, but it may not be geodesic convex under the Wasserstein metric, because we do not have convexity assumptions to the payoff function K. Though, it does not influence the convergence of the dynamics. And the convergence result is no worse than that established in [JKO].
> > >
> > > If geodesic convexity can be established (e.g. if K(x,y) is convex for any y), then we may be able to obtain better convergence rate, such as linear convergence.
> > >
> > > Based on the discussion above, the added remark 2 in the paper is incorrect. We will delete it in the next version.

---

> > > > ### Comment · Reviewer_6QW7 · 2021-11-26
> > > > **Convexity issues**
> > > >
> > > > The authors address my concerns. I will increase my score from 6 to 7.

---

> > > > > ### Author Response · Authors · 2021-11-26
> > > > > **Thanks**
> > > > >
> > > > > We thank the reviewer for the appreciation of our work and response!

---

### Official Review · Reviewer_izUn · 2021-11-01

**Correctness:** 4
**Technical Novelty And Significance:** 3
**Empirical Novelty And Significance:** 3
**Recommendation:** 6
**Confidence:** 4

**Main Review:**

1. The idea of using the different rate for the two parameter density space is implemented to solve the original min-max problem. Although the idea of having slow and fast rate is not new in the literature, the authors manage to covert the problem into a simplified problem. The reduced problem can be solved nicely.

2. The Wasserstein gradient flow is indeed a powerful tool. It is mentioned that since $F_{p,\beta}$ is strongly convex and the global convergence is expected.  Indeed, a perturbed Wasserstein Gradient flow on the density manifold is studied in [1]"Entropy dissipation via Information Gamma calculus: Non-reversible stochastic differential equations, Feng Q, Li W". It will be interesting to see that if the non-gradient flow on the density space could find its corresponding QuasiStatic Wasserstein flow story. In particular, [1] provides a faster convergence rate with perturbation added on the drift term. Is there a potential acceleration of the convergence in solving the min-max problem here?

3. The main theorem (Theorem 5) provides a weak convergence to the unique Nash equilibrium. Is it possible to achieve a stronger convergence in this QuasiStatic Wasserstein flow case? Note that the convergence of the density to the equilibrium in [1] is proved in $L^1$ norm. Instead of using the Wasserstein distance, the K-L is used instead. Can the author improve the convergence from this perspective?

4. The empirical study is only carried for 1-d torus and d-Sphere. Of course, higher dimension will be challenging. Is there anything potentially applicable or limitation of the proposed algorithms to be used in high dimension problems.

5. The analysis is conducted in compact manifolds/domains. For general global in space problems, is there a way to select the domain and then apply the algorithms in this paper to solve those general problems?

**Summary Of The Paper:**

The paper under review studies the QuasiStatic Wasserstein Gradient Flow and its application to solve two-player zero-sum games. The corresponding Langevin SDE is derived for the Fokker-Planck equation, and the discretized algorithm (GD) is proposed in this new setting.  Example of the min-max problem on 1-d torus and spheres are provided.

**Summary Of The Review:**

The proposed algorithms show its powerfulness in solving the min-max problem. Empirical study is supportive. The theoretical analysis is nicely presented with the QuasiStatic Wasserstein flow. The idea and the theoretical part both show the novelty.

---

> ### Author Response · Authors · 2021-11-22
> **Response to reviewer izUn**
>
> We thank the reviewer for providing the comments and mentioning the very interesting related work [1]. We answer the questions raised by the reviewer in the following:
>
> **1. Potential improvement based on [1] Entropy dissipation via Information Gamma calculus: Non-reversible stochastic differential equations, Feng Q, Li W.**
>
> The paper [1] provides a way to derive faster and stronger convergence results for both gradient flow and non-gradient flow in the density space. The analysis techniques developed in [1] can potentially be applied to the min-max problem studied in our paper. However, some extensions to the theories in [1] need to be made to cover the difference of the problems studied in the two works. For example, in the gradient flow case, the potential function $U$ studied in [1] is a fixed function of $x$, while in our work it depends on the distribution $p_t$. Besides, the convergence results in [1] requires the Hessian of $U$ to be positive definite. This may not hold when the object of interest is an over-parameterized GAN. We leave the exploration along this line as future work.
>
> Discussion on [1] and potential related extensions are added in the revised paper.
>
> **2. Is there anything potentially applicable or limitation of the proposed algorithms to be used in high dimension problems?**
>
> Our algorithm can be applied to high-dimensional problems. Actually, particle-based methods usually have advantages in high-dimensional problems because there is no need to use grids. To show the effectiveness of our method on high-dimensional problems, in the revised version of the paper we added one numerical experiment in which the quasistatic Langevin gradient descent method was used to train a mixed GAN on a Gaussian mixture in high-dimensional spaces. Please refer to Section 5 of the revised paper for more details.
>
>
> **3. For general global in space problems, is there a way to select the domain and then apply the algorithms in this paper to solve those general problems?**
>
> The algorithm can be directly applied to global in space problems. The problem lies in the theoretical analysis. When the domain of the problem is not compact, stronger assumptions on the problem and more sophisticated analysis are needed to show the existence of the mixed Nash-equilibrium and the convergence of the dynamics.
> For example, when the domain $\Omega$ is $\mathbb{R}^d$, if the payoff function $K$ is $0$, the min-max problem (1) in the paper becomes
> $$\min_{p\in P(\Omega)}\max_{q\in P(\Omega)} \beta^{-1}\int_\Omega p\log p dx -\beta^{-1}\int_\Omega q\log q dy.$$
> This problem does not have a well-defined Nash equilibrium, because we can always reduce the entropy by flattening the distributions.
>
> We also leave the analysis in unbounded domains as a potential future work. The reviewer can refer to the reference [Mei et al., 2018] in the paper for a mean-field analysis of neural networks in the whole space. Global in space analysis in our setting should have similar flavor.

---

### Official Review · Reviewer_4gr6 · 2021-11-02

**Correctness:** 2
**Technical Novelty And Significance:** 3
**Empirical Novelty And Significance:** 3
**Recommendation:** 6
**Confidence:** 3

**Main Review:**

The intuition and novelty of this paper is good, the authors proposes on analyzing a special setting of finding a mixed Nash equilibrium, by first observing the convergent behavior of the limiting dynamics, and then design a new algorithm via a reasonable discretization. However, the presentation and organization of this paper largely hinders the understanding of the primary contributions.

In the Introductory part of this paper, when the authors argues that the main contribution of their paper includes a convergence result of quasi static Wasserstein gradient flow dynamics under mild assumptions, and a practical discretization of the dynamics, it is insufficient to illustrate the significance of this paper. The readers lack information on whether similar results have already been obtained by previous works. Is this paper the first approach of this kind? How to compare the convergence result of this paper with [Hsieh et al. 2019], [Domingo-Enrich et al.], etc.? What are the main technical difficulties of designing such an algorithm? The related works in this paper also seems incomplete, the connection of the mean-field studies with the theoretical studies in GANs is missing. All these drawbacks have made this work hard to evaluate.

I searched for the proof of Theorems, Propositions and Lemmas in the Appendix, it seems that the proof of Theorem 5, which is the main theorem, is missing? Most of the results in the main text are direct applications of previous results, it might be better to explain the intuition of the derivation, rather than listing every steps as Lemmas, Theorems, Propositions.

**Summary Of The Paper:**

This paper focuses on the methodology of finding mixed Nash equilibrium. The authors first introduces the corresponding quasi static Wasserstein gradient flow dynamics for solving the problem, which is a limiting dynamics with $q$ seen as infinitely faster than $p$. Then the continuous dynamics is proved to be convergent under mild assumptions. Furthermore, by discretizing the above gradient flow, the authors proposes a practical algorithm for solving min-max optimization problems with GAN as a special example. Experimental results shows the superiority over the vanilla LGDA algorithm especially with large $\beta$’s.


**Summary Of The Review:**

Overall, this paper provides a novel algorithm towards finding the mixed Nash equilibrium. However, the presentation of this paper does not support their result well. As a result, the paper is not ready for acceptance at this time.


----------
Post Rebuttal Comments:

The authors have addressed my concerns. Therefore I'm increasing my score from 5 to 6.

---

> ### Author Response · Authors · 2021-11-22
> **Response to reviewer 4gr6**
>
> We thank the reviewer for the appreciation of the novelty of our work. To address the reviewer's questions, we improved the presentation of our paper in the following ways: (1) More comparison with previous works are added; (2) Related works part is made more integrated, with the connection of our work with the mean-field studies added; (3) More explanations are added to the theoretical part. Please see detailed responses below:
>
> **1. The readers lack information on whether similar results have already been obtained by previous works. Is this paper the first approach of this kind?**
>
> This paper is the first work that explores the quasistatic Wasserstein dynamics for finding mixed Nash equilibrium. Theoretically, we show the global convergence of the quasistatic dynamics. In practice, we design a quasistatic Langevin gradient descent which can be applied to the training of mixed GANs. Both contributions mentioned above do not exist in previous works. The problem of finding mixed Nash equilibrium for GANs has been studied by some other works. However, all those works deal with different dynamics and different algorithms. Some comparisons will be detailed under question 2.
>
> **2. How to compare the convergence result of this paper with [Hsieh et al. 2019], [Domingo-Enrich et al.], etc.?**
>
> **[Hsieh et al. 2019]:** In this work, the authors studied a mirror descent algorithm for finding mixed Nash equilibrium. The algorithm is first written as an iteration of continuous probability distributions. Convergence results on the level of continuous distributions is then obtained. However, since the mirror descent does not consider gradient under the Wasserstein metric, discretizing the algorithm to particle level and making it practical is highly nontrivial. Sampling by SGLD is conducted in every iteration, causing big computational cost. Heuristics without theoretical justification are used to reduce the computational cost. As a comparison, we study the dynamics under Wasserstein metric, hence the discretization of our QSWGF is naturally a SGLD-like algorithm. The algorithm design procedure in our work is more natural.
>
> **[Domingo-Enrich et al.]:** In this work, the authors studied a Wasserstein gradient descent-ascent flow, where the speeds of the two distributions are the same. In our quasistatic dynamics, however, one distribution is made infinitely faster than the other, so that it is considered to be at equilibrium at any time. The advantage of the quasistatic dynamics mainly lie in the theoretical study. We can show global convergence of the quasistatic dynamics, while in [Domingo-Enrich et al.] the authors can only show the convergence of the time average of the dynamics' trajectory.
>
> **3. What are the main technical difficulties of designing such an algorithm?**
>
> The mixed Nash equilibrium problem is a problem for probability distributions. However, finally any practical algorithms should treat finite number of variables instead of continuous probability distributions. Hence, the main difficulty for this problem is to find a dynamics for probability distributions that is both theoretically solid and easy to implement as an algorithm of discrete variables. The QSWGF meets the two requirements simultaneously.
>
> **4. The related works in this paper also seems incomplete, the connection of the mean-field studies with the theoretical studies in GANs is missing.**
>
> In the mean-field analysis of neural networks, neural networks are parameterized by a probability distribution, and the training dynamics is a gradient descent flow under the Wasserstein metric. Hence, Wasserstein gradient flow is extensively analyzed in the mean-field studies for neural networks. In our work, we also study a problem involving probability distributions, and a dynamics under the Wasserstein metric. Therefore, the study is related with the mean-field analysis of neural networks, and the theoretical techniques therein can be borrowed by our work. This connection is also drew in [Domingo-Enrich et al.] (Section 2). We have made this connection clearer in the revised related works section.
>
> **5. It seems that the proof of Theorem 5, which is the main theorem, is missing?**
>
> We have a complete and rigorous proof for Theorem 5. Due to the space limit, we give a proof sketch in the main text and put the full proof into the appendix. The proof sketch is right after the statement of Theorem 5. the logic of the whole proof is presented, and important intermediate results are summarized as lemmas in a logical order. In the revised paper, more intuitive explanations are added in the proof sketch.
>
> **6. Most of the results in the main text are direct applications of previous results.**
>
> We are doing extensions to previous results rather than direct applications. For example, in the proof of Lemma 1 and Lemma 2 (which are central to the main theorem), we extend previous analysis to a new energy $\log Z$, which is never dealt with in previous works.

---

### Author Response · Authors · 2021-11-22
**Paper updated**

We thank all the reviewers for the valuable comments and suggestions. We have made careful revision to the paper according to the reviews. Specifically, the following major changes are made:

1. The discussion of related works is improved. Careful comparison is made with important related papers, highlighting the difference of our work with others. Some missing related references are added and discussed.
2. More intuitive explanations are added to the theoretical analysis, especially in the proof sketch of the main theorem. We hope this can help readers better understand the our analysis.
3. A numerical experiment is added to show the effectiveness of our quasistatic Langevin gradient descent method on high dimensional problems. In this experiment, we train mixed GANs to learn Gaussian mixtures in high dimensional spaces.

For details of the revision, please refer to the new version of the paper. All changes (except correction of typos and grammar problems) are marked blue.

---

### Decision · Program_Chairs · 2022-01-20

**Decision:**

Accept (Poster)

**Comment:**

The authors first consider a mean field two player zero sum game and consider quasistatic Wasserstein gradient flow dynamics for solving the problem. The dynamics is proved to be convergent under some assumptions. Finally, the authors provide a discretization of the gradient flow and using this proposes an algorithm for solving min-max optimization problems. They use this algorithm for GAN's as the main example. Experimental results claim that the algorithm outperforms langevin gradient descent especially in high dimensionas.

This paper sits right at the border. But subsequent to the author response, one of the reviewers has updated the score and seems more positive about the paper. In view of this, I am leaning towards  an accept.